# Moyamoya Vasculopathy in Neurofibromatosis Type 1 Pediatric Patients: The Role of Rare Variants of *RNF213*

**DOI:** 10.3390/cancers15061916

**Published:** 2023-03-22

**Authors:** Marzia Ognibene, Marcello Scala, Michele Iacomino, Irene Schiavetti, Francesca Madia, Monica Traverso, Sara Guerrisi, Marco Di Duca, Francesco Caroli, Simona Baldassari, Barbara Tappino, Ferruccio Romano, Paolo Uva, Diego Vozzi, Cristina Chelleri, Gianluca Piatelli, Maria Cristina Diana, Federico Zara, Valeria Capra, Marco Pavanello, Patrizia De Marco

**Affiliations:** 1U.O.C. Genetica Medica, IRCCS Istituto Giannina Gaslini, 16147 Genova, Italy; 2Department of Neurosciences, Rehabilitation, Ophthalmology, Genetics, Maternal and Child Health, Università Degli Studi di Genova, 16145 Genova, Italy; 3Dipartimento di Scienze della Salute, Università di Genova, 16132 Genova, Italy; 4U.O.C. Neurologia Pediatrica e Malattie Muscolari, IRCCS Istituto Giannina Gaslini, 16147 Genova, Italy; 5LABSIEM (Laboratory for the Study of Inborn Errors of Metabolism), IRCCS Istituto Giannina Gaslini, 16147 Genova, Italy; 6U.O.C. Genomica e Genetica Clinica, IRCCS Istituto Giannina Gaslini, 16147 Genova, Italy; 7Unità di Bioinformatica Clinica, Direzione Scientifica, IRCCS Istituto Giannina Gaslini, 16147 Genova, Italy; 8Genomic Facility, Istituto Italiano di Tecnologia, 16163 Genova, Italy; 9U.O.C. Neurochirurgia, IRCCS Istituto Giannina Gaslini, 16147 Genova, Italy

**Keywords:** neurofibromatosis type 1, Moyamoya, next-generation sequencing (NGS), *NF1*, *RNF213*

## Abstract

**Simple Summary:**

Neurofibromatosis type 1 and Moyamoya disease can be associated in Moyamoya syndrome (MMS), which causes cerebral arteriopathy consisting of a progressive steno-occlusion of the intracranial arteries. Here, we investigated if rare variants of *RNF213*, the most common genetic risk factor for Moyamoya, could act as genetic modifiers of MMS phenotype in a pediatric cohort of patients. Next-generation sequencing of *NF1* and *RNF213* genes was performed on genomic DNA extracted from patients’ blood. We found that in MMS patients, *RNF213* does not appear to modify Moyamoya occurrence risk. Rather, it is more probable that the loss of neurofibromin 1, the protein encoded by the *NF1* gene, is responsible by itself for the excessive proliferation of vascular smooth muscle cells causing arterial stenosis. Further studies in larger cohorts of patients are necessary to confirm these findings and to identify other genetic factors in order to increase our understanding of MMS pathogenesis.

**Abstract:**

Neurofibromatosis type 1 (NF1) is a neurocutaneous disorder caused by mutations in *NF1* gene, coding for neurofibromin 1. NF1 can be associated with Moyamoya disease (MMD), and this association, typical of paediatric patients, is referred to as Moyamoya syndrome (MMS). MMD is a cerebral arteriopathy characterized by the occlusion of intracranial arteries and collateral vessel formation, which increase the risk of ischemic and hemorrhagic events. *RNF213* gene mutations have been associated with MMD, so we investigated whether rare variants of *RNF213* could act as genetic modifiers of MMS phenotype in a pediatric cohort of 20 MMS children, 25 children affected by isolated MMD and 47 affected only by isolated NF1. By next-generation re-sequencing (NGS) of patients’ DNA and gene burden tests, we found that *RNF213* seems to play a role only for MMD occurrence, while it does not appear to be involved in the increased risk of Moyamoya for MMS patients. We postulated that the loss of neurofibromin 1 can be enough for the excessive proliferation of vascular smooth muscle cells, causing Moyamoya arteriopathy associated with NF1. Further studies will be crucial to support these findings and to elucidate the possible role of other genes, enhancing our knowledge about pathogenesis and treatment of MMS.

## 1. Introduction

Neurofibromatosis type 1 (NF1; OMIM #162200) is an autosomal dominant neurocutaneous disorder impacting one in 3000 children and affecting males and females of all ethnic groups equally. NF1 is caused by mutations in *NF1 (chromosome 17q11.2),* the gene coding for neurofibromin 1, a negative regulator of the Ras signal transduction pathway that has a role in both tumor suppression and the regulation of cell growth and proliferation [1]. NF1 is characterized by prominent skin features (hyperpigmented patches termed café-au-lait spots and neurofibromas), optic tumors and other central nervous system tumors, bony anomalies, cognitive impairment and an increased risk of certain non-nervous system cancers [2].

Cerebral arteriopathy in NF1 patients is estimated to occur in 2.5% to 6% of children [3,4], and the most prevalent form of NF1-associated arteriopathy is Moyamoya disease (MMD). MMD consists of a progressive steno-occlusion of the intracranial arteries resulting in collateral vessel formation that causes a characteristic “puff of smoke” (translation of the Japanese term moyamoya) appearance in cerebral angiography [5]. MMD is considered the major cause of ischaemic and haemorrhagic stroke, with an annual incidence of 0.5–1.5 per 100,000 individuals in Asian countries but only 0.1 per 100,000 in other regions [6,7]. When MMD is present in association with other specific diseases, such as Down syndrome, sickle cell disease, atherosclerosis or NF1, it is referred to as Moyamoya syndrome (MMS) and, in association with NF1 in particular, it is called MMS type 1 [8,9,10]. NF1 and MMD are two distinct genetic conditions that can sometimes co-occur in the same individual, but the relationship between them and their role in MMS is not fully understood. While NF1 is not a direct cause of MMS, individuals with NF1 may have an increased risk of developing MMD. In fact, it is estimated that approximately 2–6% of individuals with NF1 may develop MMD or MMS [11,12].

MMD symptoms can vary in severity, ranging from transient ischemic attacks to persistent neurological deficits such as motor impairment, speech, sensory and consciousness disturbances, headache, movement disorders and convulsions [13]. Individuals displaying NF1 should be closely monitored for the possible development of MMD and MMS, and prompt treatment should be initiated if these conditions are detected. The management of patients with MMS can be very complex and its treatment strategy inclu. Please des surgical revascularization in selected cases [14,15,16]. Clinical progression and follow-up neuroimaging evaluation play a crucial role in the selection of the best therapeutic approach [14,15,16,17].

MMD aetiology has long remained elusive. However, in 2011, the *RNF213* gene (chromosome 17q25.3), which encodes a protein named mysterin, was identified as a susceptibility gene for MMD [18,19]. Mysterin is a huge protein with ATPasic and ubiquitin–ligase activities, performed by its AAA+ and RING finger domains, respectively.

The Arg4810Lys variant in *RNF213* is associated with an increased risk for MMD in Asiatic populations with a founder effect. However, its penetrance (ratio of affected individuals among carriers) is lower than 1%, suggesting a synergistic relationship with additional environmental and genetic risk factors [20,21,22]. This mutation has been shown to disrupt the normal function of mysterin, which is involved in maintaining the structure and function of blood vessels in the brain [11]. Other genetic rare variants, localized in the C-terminal region encompassing the RING finger domain of *RNF213*, have been associated with MMD in Caucasians, suggesting that the genetic basis of MMD may be diverse. Of note, a splicing variant in *RNF213* was recently reported in a patient with *SPRED1*-related Legius syndrome and MMS, further supporting the association between *RNF213* and a predisposition to moyamoya vasculopathy [23]. It is worth noting that not all individuals with MMD have mutations in the *RNF213* gene, and other genetic and environmental factors may also play a role in the development of this disorder. Other genes, including *ACTA2*, *GUCY1A3*, and *MYH11*, have been associated with MMD [24,25,26]. However, the identification of *RNF213* mutations has provided valuable insights into the pathogenesis of MMD and may eventually lead to the development of new treatments for this condition.

Although MMD occurs frequently among Asian populations and the prevalence of NF1 is not correlated with ethnicity, the majority of MMS patients are reported in Europe and the USA, demonstrating that different genetic backgrounds among ethnic groups may play a role [27,28].

So far, the genetic factors predisposing patients to MMS have not been identified. Neurofibromin 1 is expressed in the mammalian endothelium and vasculopathy in NF1 patient, resulting from excessive angiogenesis through the proliferation of endothelial cells and pericytes [29,30]. It has been shown that the loss of neurofibromin in human endothelial cells increases migration and proliferation in response to growth factors via hyperactivation of the Ras pathway, which could contribute to the development of arterial stenosis (narrowing of the arteries) and other vascular pathologies [29]. However, it is not clear whether the loss of *NF1* in endothelial cells alone is sufficient to cause the excessive proliferation of vascular smooth muscle cells, which are the cells that contribute to arterial stenosis [31,32]. Vascular smooth muscle cell proliferation is a complex process that can be influenced by many factors, including growth factors, inflammatory signals, and mechanical stresses [32,33]. While the loss of *NF1* in endothelial cells may contribute to the development of arterial stenosis by altering blood vessel development and function, it is likely that additional factors are involved in the excessive proliferation of vascular smooth muscle cells. Further research is needed to fully understand the role of *NF1* in arterial stenosis and other vascular pathologies.

In this research, we investigated whether rare variants in the *RNF213* gene may act as genetic modifiers of the MMS phenotype in a cohort of pediatric patients. The study included 20 Caucasian patients with MMS, 25 either Caucasian or Asian/African patients with isolated MMD, and 47 children with NF1 but without signs of MMD. Next-generation sequencing (NGS) of *NF1* and *RNF213* genes was performed on genomic DNA extracted from patients’ blood. We found that *RNF213* rare variants may only play a role in a minority of pediatric MMD patients. Conversely, the *RNF213* gene does not appear to contribute to the Moyamoya occurrence risk in subjects with MMS. However, it is possible that the loss of neurofibromin 1 in endothelial cells could be responsible itself for the excessive proliferation of vascular smooth muscle cells leading to arterial stenosis.

## 2. Materials and Methods

### 2.1. Patients and Controls

Ninety-two unrelated probands were enrolled, 94% of whom had Caucasian ancestry. The non-Caucasian patients were of Asian (1), African (2), or Hispanic (2) ancestry. Sixty-six children were referred to the Department of Neurology and Neurosurgery at IRCCS G. Gaslini (Genoa, Italy), while 26 were enrolled by other Italian hospitals (Ancona, Palermo, Pavia, Torino) and sent to the Medical Genetics Department of IRCCS G. Gaslini for molecular diagnostic purposes between 2009 and 2022. Sixty-seven probands were affected by NF1. Among them, 47 did not show signs of Moyamoya arteriopathy after at least 3 years of follow-up (NF1, control group), while 20 already had the arteriopathy at the time of diagnosis or developed it during the course of NF1 follow-up (MMS, study group). NF1 was diagnosed if the patients had at least two or more NIH diagnostic criteria [34] and harbored a loss-of-function mutation in *NF1*. Clinical and imaging data of patients with isolated or syndromic Neurofibromatosis were extracted from an internal database of almost 600 patients, described in Scala et al. (2021) [35]. As a second control group, we recruited 25 children with isolated childhood-onset Moyamoya disease (MMD, control group). Diagnosis of Moyamoya was performed based on brain MRI or cerebral angiography showing a stenosis of the terminal part of the terminal ICA and/or the proximal portions of the anterior and/or middle cerebral arteries associated with abnormal vascular collaterals. Patients were recruited who had been diagnosed with both unilateral and bilateral stenosis. All children presenting with ischemic stroke symptoms underwent surgical indirect bypass revascularization. Patients with radiation-induced MMD were excluded. In addition to these 92 probands, 34 affected or clinically healthy relatives were included for the genotyping of candidate variants. All patients and their participating relatives provided written informed consent for genetic analysis. The study was conducted in accordance with the ethical standards of the declaration of Helsinki.

### 2.2. DNA Extraction

Genomic DNA was extracted from peripheral venous blood samples using the commercial kit Qiasymphony DNA Midi (Qiagen, Hilden, Germany) following the manifacturer’s instructions. DNA concentration and quality was evaluated by both Nanodrop 2000 spectrophotometer (Thermo Fisher Scientific, Walthman, MA, USA) and Qubit^®^ 3.0 Fluorometer (Thermo Fisher Scientific) using the Qubit™ dsDNA HS Assay Kit (Thermo Fisher Scientific).

### 2.3. Next-Generation Sequencing (NGS)

Two Ion AmpliSeq NGS on-demand panels were created using the Ion AmpliSeq Designer tool accessed on 14 September 2018 (http://www.ampliseq.com) by Thermo Fisher Scientific for the sequencing of the entire coding region and 10 bp of the adjacent intronic regions of *RNF213* (NM_001256071.2) and *NF1* (NM_000267.3). The target regions were entered into the online tool and the resulting 123 amplicons, size 125–375 bp, for *RNF213* (IAD) and 120 amplicons for *NF1* (IAD), size 125–275 bp, were divided into two primer pools. Libraries were prepared, starting from 10 ng of genomic DNA, using the AmpliSeq Library Kit 2.0 (Thermo Fisher Scientific) according to the manufacturer’s instructions and barcoded with the Ion Xpress Barcode Adapter. The final concentration was evaluated with a Qubit^®^ 3.0 Fluorometer using the Agilent High Sensitivity DNA Kit. Libraries were then diluted to 100 pM and pooled together. Template preparation and chip loading were performed on the Ion Chef System (Thermo Fisher Scientific). The sequencing was performed on the Gene Studio S5 (Thermo Fisher Scientific.) using 510 Ion Chips. Fastq were analyzed with Ion Reporter pipeline v.5.6 (Thermo Fisher Scientific) and by CLC Genomics Workbench 6.5.1 software (Qiagen) for filtering out poor-quality reads, alignment on GRCh37/hg19 reference, variant detection, and coverage analysis. Variants were considered of higher priority if: (a) they were predicted to affect protein-coding sequences (including synonymous, nonsynonymous, frameshift deletion, and stop-gain variants in exonic or splicing regions); (b) they showed a low frequency in reference databases (minor allele frequency was less than 0.01) based on the data of the Genome Aggregation database (gnomAD v2.1.1) and the in-house exome controls dataset, Biogear. Biogear includes exome sequencing data from 2504 healthy individuals, mainly of Caucasian origin, who are unaffected parents of patients with neurodevelopmental disorders. We checked for the of variants presence in the ClinVar (http://www.ncbi.nlm.nih.gov/clinvar/ (accessed on 1 December 2022) and Leiden Open Variation Database (LOVD) databases and variants were classified according to ACMG criteria using the Franklin web tool. Only pathogenic (P)/likely pathogenic (LP)/uncertain significance (VUS)/conflicting for ClinVar criteria were prioritized.

The variants were validated by Sanger sequencing, using High-Fidelity Platinum Master Mix (Thermo Fisher Scientific) for PCR amplification and the BigDye Terminator v1.1 kit (Thermo Fisher Scientific) for sequencing.

### 2.4. Multiplex Ligation-Dependent Probe Amplification (MLPA)

MLPA was performed in order to detect exonic deletions of *NF1* using two commercial kits, namely the SALSA P081/P082 kits (MRC-Holland, Amsterdam, The Netherlands). We used 100 ng of denatured genomic DNA extracted from patients and control individuals in the overnight annealing of the exon-specific probes and the subsequent ligation reaction. PCR was carried out with FAM-labelled primers using 10 μL of probe ligation product. Separation was performed using an ABI Prism 3100 Genetic Analyzer (Thermo Fisher Scientific) and MLPA data analysis was performed with the Coffalyser.Net™ v.1 software (MRC-Holland). A reduction in the peak area values to <0.5 or their increase to >1.5 was considered an indication of a deletion or a duplication, respectively.

### 2.5. RNA Extraction, RT-PCR and Sequencing of NF1 Gene

In order to detect variants affecting the alternative splicing of *NF1*, we performed re-sequencing of the *NF1* gene, starting from the RNA in some patients which resulted negative from NGS and MLPA analysis. Peripheral blood was collected in PAXgene Blood RNA tubes and subjected to total RNA extraction using the PAXgene Blood RNA kit (Qiagen) following the manufacturer’s instructions. RNA amount and quality were evaluated with NanoDrop. Total mRNA was retro-transcribed onto cDNA using the SuperScript First-Strand Synthesis System kit (Thermo Fisher Scientific). Twenty-four overlapping RT-PCR primer sets, generating products of sizes ranging between 319 and 621 bp, were designed to cover the *NF1* coding sequence of about 8.6 kb in length (NM 000267.3) and they are presented in the “Appendix A”. PCR reactions were carried out with the Platinum PCR SuperMix High Fidelity kit (Thermo Fisher Scientific), using 2 µL samples of cDNA as templates in a total volume of 50 μL. DNA amplification was achieved by the use of an initial denaturation step at 94 °C for 5 min, followed by 35 cycles of 94 °C for 15 s, 60 °C for 30 s, and 68 °C for 1 min.

### 2.6. Statistical Analysis

The baseline characteristics of patients were summarized with number and percentages for categorical variables, and with mean and standard deviation and median with range for continuous variables. The enrichment of putatively damaging *RNF213* variants was tested in cases vs. controls. *RNF213* loss-of-function variants (>stop-loss, stop-gain, splice site variants, frameshift) and missense variants, which were predicted to be pathogenic by both SIFT and Polyphen, with MAF below 0.01, in two control datasets (gnomAD v.2.1.1 and Biogear) were annotated. The association was then tested by comparing the number of cases and controls carrying at least one such variant in the *RNF213* gene to avoid type-one error inflation due to a linkage disequilibrium between variants. Comparison of two rates (sample vs. healthy databases) was expressed as the ratio of the two rates (IRR: incidence rate ratio), with its 95% confidence interval (calculated with exact Poisson method) and associated *p* value. If the *p* value was less than 0.05, it was concluded that the two rates were statistically significantly different.

## 3. Results

Basic characteristics of the patients enrolled in the study are shown in Table 1.

The female/male sex ratio of the whole study group was 1.0 and 1.5 for both MMS and MMD groups, but we did not find a significant predominance of females among the groups (*p* = 0.34). The total age of enrolled the subjects ranged from 1 to 18 years (mean age 9.3 years). Overall, we found statistical differences in the age (*p* = 0.036) and in the age at diagnosis (*p* = 0.003) between groups. Pair-wise comparison showed that the mean age of MMD group was significantly lower (7.4 years; *p* = 0.032), as was the age at the diagnosis (4.6 years, *p* = 0.002), with respect to the MMS group. No patients had a familial Moyamoya history, while 2 patients of the MMS group and 4 of the NF1 group had familial forms of Neurofibromatosis.

To confirm the results of the NF1 clinical diagnosis, *NF1* re-sequencing was performed in 20 patients with MMS, as well as in 47 probands with isolated NF1. A total of 19 out of 20 (95%) MMS probands were carriers of P/LP loss-of-function (4 missense, 3 stop-gain, 7 frameshift, 2 splice sites, 2 exonic deletions) variants in the *NF1* gene, with the exception of proband #14, which responded negatively to mutational analysis. However, we could not search for variants causing exon skipping or exonic *NF1* deletion since the family refused further analysis. Moreover, patient #18 was a familial case of NF1 and carried a VUS in *NF1*, located at +4 bp at the intron–exon boundary. This variant was inherited from the affected father and also segregated in the affected brother; thus, it is likely that *NF1* c.586+4dupA has a pathogenic role, even if functional studies of the variant are necessary to draw definitive conclusions (Table 2; Figure 1).

All children of NF1 group were carriers of P/LP loss-of-function variants of *NF1* gene (7 missense, 17 frameshift, 1 in frame deletion, 15 stop-gain, and 7 splice site variants) (Table 3; Figure 2).

All the identified *NF1* variants are absent in public databases and are either absent or present at very low frequencies (MAF < 1:10.000) in our exome control database; among them, 41 were reported in Clinvar. Thus, the clinical suspicion of NF1 was confirmed by genetic testing, both in isolated and syndromic patients.

To evaluate the role of genetic factors other than *NF1* in the onset of Moyamoya in Neurofibromatosis patients, we re-sequenced the entire coding region, including the intron–exon boundaries of *RNF213*, the major susceptibility gene for MMD in the three groups of patients. Of note, the *RNF213* p.(R4810K) variant was absent in all groups of patients since they were mostly of Caucasian origin. A total of 18 *RNF213* rare variants (MAF < 0.01) were identified (Table 2, Table 3 and Table 4; Figure 1, Figure 2 and Figure 3).

Among these, 3 were synonymous and 3 were missense variants that are predicted to be benign and, thus, they did not fulfil the criteria for inclusion in the association tests. Among the remaining 12 rare variants, 4 were novel and 8 were present with very low frequency in control databases and all were predicted to be VUS or LP, according to the ACMG guidelines. The p.(Pro1721Leu) was shared by two unrelated probands and, interestingly, 6 variants were just reported in patients with Moyamoya (p.(Pro1721Leu); p.(Gln2184Arg); p.(Ile3318Val); p.(Cys4017Ser); p.(Ala4399Thr); p.(Ala5021Val)) [36,37]. Three MMS patients (15%) were carriers of missense variants in *RNF213*, the p.(Ala1241Thr), p.(Thr1626Met), and p.(Ala3468Val) variants, all of which were classified as VUS; however, these variants were located in the N-terminal region of the protein encoded by *RNF213* and were outside of the AAA+ and RING finger domains. Moreover, the MMS patient who carried the *RNF213* p.(Ala1241Thr) variant was the only child in whom no variants could be detected through the mutational screening of the *NF1* gene (Table 2, Figure 1). In the MMD group, we identified 9 (36%) variants (1 stop-gain and 8 missense) of uncertain or likely pathogenic significance. More notably, 4 missense variants clustered in C-terminal region of the RNF213 protein where the variants associated with Moyamoya also clustered. Additionally, 2 variants, the p.(Cys4017Ser) and p.(Cys4035Arg) variants, were located in the RING finger domain. Segregation analysis among available parents showed that *RNF213* variants were inherited from unaffected parents and were low-penetrance variants, with the exception of the p.(Cys4017Ser) and p.(Cys4035Arg) in children with isolated Moyamoya vasculopathy (Table 4, Figure 3). Only 1 in 47 (2%) of the NF1 patients was a carrier of the *RNF213* p.(Pro1721Leu) variant classified as a VUS, but it was located far from the C-terminal region of RNF213 protein (Table 3, Figure 2). This variant had previously been published in a series of MMD probands and controls of European ancestry [36]. We cannot exclude that the *RNF213* p.(Pro1721Leu) variant, also identified in a subject with MMD proband, may be a risk factor in this NF1 patient.

To evaluate if the enrichment of putatively damaging R*NF213* rare variants may act as a genetic modifier of the Moyamoya risk in our patients, we performed a gene-based burden testing approach using two control datasets: the publicly available gnomAD.v2 database and the in-house Biogear dataset (Table 5).

The number of probands and controls harboring rare putatively damaging variants (P/LP/VUS) of *RNF213* were annotated and stratified into carriers of rare variants along the 3 functional domains of the RNF213 protein: (1) N-terminal (aa 1–3996), (2) RING Finger domain (aa 3997–4035), and (3) C-terminal (aa 4036–5207). None of the probands and controls was homozygous for any mutated allele. We did not identify a positive trend for the accumulation of such variants among MMS patients, since only 15% of them were carriers of putatively damaging *RNF213* variants, localized to the N-terminal region with a frequency similar to that found in the Biogear dataset (11%; *p* = 0.58). Conversely, the comparison between MMD probands and controls showed an enrichment of rare *RNF213* variants located in the RING finger domain and in the C-terminal region in MMD cases (*p* < 0.004 for comparison with mutated Biogear dataset and *p ≤* 0.001 for comparison with mutated gnomAD controls). Of note, these associations also remained significant when we considered the two domains as a unique hot spot region (RF + C-term; *p* < 0.001). The IRR values were very high, especially for MMD probands carrying variants in the RING finger domain, suggesting that the sub-region of the gene encoding this domain is the most intolerant to variations. The *RNF213* p.(Cys4017Ser) and p.(Cys4035Arg) identified in two children with isolated MMD occurred de novo, were absent in controls, and affected two out of seven highly conserved key cysteine residues within the RING finger domain, which mediates Zn binding. Cys4017 is involved in the first Zn coordination site, while Cys4035 is the last residue of the domain and participates to the second coordination site (Figure 4).

Both variants are predicted to have an impact on the 3D structure and functional activity of the RING finger domain. Variants located in the N-terminal region did not seem to play a role in the risk of MMD probands (*p* = 0.85 vs. Biogear controls). Overall, these results revealed that rare variants of the *RNF213* gene only impinge on Moyamoya risk for the MMD group.

## 4. Discussion

The association between NF1 and MMD is typical of the pediatric age and affects a minority of children with NF1 [3,4]. While many patients are asymptomatic, MMS needs early diagnosis and appropriate treatment to reduce the incidence of ischemic events [14,15,16,17]. The identification of risk factors represents an important problem in the clinical management of this minority of patients [15,16,17]. To date, *RNF213*, mapping onto 17q25, is the only gene associated with MMD, both in Asiatic and Caucasian populations, and its proximity to *NF1*, on 17q11.2, has been evoked to explain the association between the two diseases. However, *NF1* and *RNF213* participate in different pathways. *NF1* is a negative regulator of the Ras pathway that controls cell division and proliferation [38], while *RNF213* inhibits the non-canonical Wnt signaling pathway, promoting vessel regression [39].

Phi et al. found that the *RNF213* p.R4810K variant represents an important risk factor for MMS occurrence in Asiatic patients with a calculated crude OR of 50.57 (95% CI 1.57–1624) [40]. However, no study has investigated the genetic basis of MMS in Caucasian cohorts since the one published by Santoro et al. [28]. In this study, the authors demonstrated a possible pathogenic role for *NF1* alone. However, they did not exclude, given the rarity of MMS events, that other genetic factors could be implicated.

In our study, we explored if *RNF213* may play a role in the pathogenesis of MMS by re-sequencing the entire coding region in a cohort of pediatric MMS patients with genetically confirmed NF1 diagnosis.

We showed that *NF1* loss-of-function mutations were identified in all, except one patient, with isolated NF1 or syndromic forms (MMS), confirming the clinical diagnosis in patients who met clinical inclusion criteria. By re-sequencing *RNF213*, we confirmed that the Asian founder mutation in *RNF213*, p.R4810K, is not present in patients of European descent. We found a low rate of putatively damaging rare variants of MMS (15%) and NF1 study groups (2%). Of note, all MMS variants were located outside of the C-terminal region of *RNF213* and were reported in our in house controls dataset and/or in population databases. Burden tests identified no increased risk for MMS patients who were carriers of *RNF213* rare variants in the N-term domain as well as in RF and C-term regions. Thus, the *RNF213* gene did not seem to play a role for Moyamoya risk in pediatric MMS patients. On the contrary, among MMD probands, 12 out 25 (48%) were carriers of rare *RNF213* nonsynonymous variants. The proportion of mutants among MMD group is similar to that (52%) reported by Guey et al. in a cohort of childhood-onset MMD probands of Caucasian ancestry [36]. When we restricted the analysis to the putatively damaging variants of *RNF213*, gene burden testing identified a positive association between *RNF213* missense variants and MMD. The association was only found for those variants encompassing the C-terminal region of *RNF213* and was even stronger for variants located in the RING finger domain, demonstrating that MMD-related variants are significantly grouped in a C-terminal region that includes the RING finger domain of *RNF213*. Interestingly, p (Cys4017Ser) and p.(Cys4035Arg), which are absent in control databases and affect highly conserved residues, were found to occur de novo in two patients with severe and early-onset cases of MMD. Gene burden testing showed significant results using the gnomAD database that includes multiethnic individuals, thus generating a spurious association due to population stratification bias. However, the results remain significant all the same when we used for comparisons the in-house Biogear dataset, which catalogs variations from healthy ancestry-matched individuals. A limitation of the burden analysis is the different sequencing strategy (NGS) of cases and database controls (exome sequencing). Additionally, the different pipeline used to capture the variants might generate false-positive associations. Thus, while in East Asian countries MMD is strongly associated with the p.R4810K variant in the *RNF213* gene that segregates with disease in an autosomal dominant manner with reduced penetrance [37], in accordance with findings of Guey et al. (2017) [36], our results provided evidence that several heterozygous rare *RNF213* variants of a non-p.R4810K variety which are localized in the C-terminal hot spot domain, play a role for a substantial proportion of non-syndromic Moyamoya patients of European ancestry.

Given that the *RNF213* gene did not seem to play a role for Moyamoya risk in pediatric MMS patients, we could not exclude that other genetic factors may be involved. This possibility will be clarified by exome sequencing of MMS probands, as has just been reported by Santoro et al. (2018), who identified *MRVI1* as a susceptibility gene in a large Italian family [41].

This lack of association between *RNF213* variants and MMS strengthened our suspicion that loss of neurofibromin, a negative regulator of Ras activity which is highly expressed in endothelial and smooth muscle cells of vessels, may be the underlying cause of the vascular manifestations in NF1, including MMS. Indeed, *Nf1* knock-out (ko) mice show exaggerated angiogenic responses, which caused growth factors-mediated Ras hyper-activation and increased signaling in endothelial cells. Ras activation in primary endothelial cells results in the perturbation of the endothelial cell vasculogenic program and in a pro-survival and pro-proliferative phenotype that disrupts normal vascular morphogenesis [42]. Moreover, the in vitro sensitivity of the Nf1-ko endothelial cell proliferation and the normalization of the vascular morphogenesis following treatment with low-dose rapamycin suggests that anti-tumor drugs may be effective for treating certain types of vascular dysfunctions in NF1 patients [43,44].

Based on the available literature and the findings from our current study, it is possible to speculate that MMS may be a digenic disease caused by mutations in both *RNF213* and *NF1*. Alternatively, it is possible that the loss of function of *NF1* alone may itself predispose patients to an increased risk for vascular complications through the dysregulation of cell function within the vascular endothelium. Finally, we cannot exclude the possibility that development of Moyamoya vasculopathy in heterozygous human NF1 patients might arise from occasional “second hit” somatic mutations of the normal *NF1* allele in a critical tissue or that it could be the consequence of other genetic and environmental factors that interact with neurofibromin in maintaining brain vessel integrity.

In a diagnostic context, our study raises the question of the putative benefit of a genetic screening of *RNF213* in MMS patients. In fact, MMD is a progressive disease and the prevention of strokes and the resulting co-morbidities depends on the early identification of at-risk individuals. In general, there is no consensus about the management of NF1-associated vasculopathies and pediatric care guidelines do not recommend neuroimaging as a surveillance modality [34]. Thus, the identification of noninvasive prognostic factors is crucial for the management of pediatric patients. The data presented here suggest that screening for rare variants of the C-terminal domain of *RNF213* should be considered only for pediatric patients with isolated MMD and not for syndromic MMS patients.

## 5. Conclusions

Our study aimed to expand our knowledge about the genotype–phenotype correlations in MMS patients, contributing to the improvement of their genetic counselling. Our analyses did not show a direct involvement of *RNF213* variants in the pathogenesis of MMS in pediatric patients. However, the sequencing of larger cohorts will be important to confirm this finding. The results of our study suggest that *RNF213* genetic testing is not particularly beneficial for the enhancement of the clinical management of MMS patients. In this regard, the identification of biomarkers for pediatric Moyamoya vasculopathy is one of the main challenges to be addressed in the next future. NF1 treatment requires multidisciplinary follow-up strategies according to the patient’s age and should include referral to specialists for the treatment of complications. Since only a minority of NF1 patients develop Moyamoya vasculopathy, vascular screening currently remains the only diagnostic tool which can be used for the early identification of MMS and other cerebral arteriopathies.

## Figures and Tables

**Figure 1 cancers-15-01916-f001:**
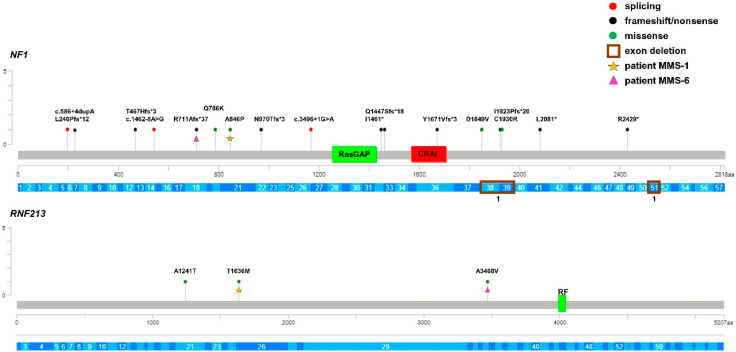
Lolliplot graph showing distribution and frequency of *NF1* and *RNF213* variants identified in MMS patients. Yellow star and violet triangle identify variants carried by the same patient. RasGAP: Ras GTPase activating protein domain; CRAL: CRAL-TRIO lipid-binding domain (cellular retinaldehyde-binding protein and TRIO guanine exchange factor); RF: RING finger domain; aa: number of protein amino acids; c.: number of nucleotides in the coding region of the gene. * stop codon. The gene exons are schematized in dark and light blue. Number 1 under the brown rectangle (exonic deletions) indicates the number of the patients who were carriers of exonic copy number variations.

**Figure 2 cancers-15-01916-f002:**
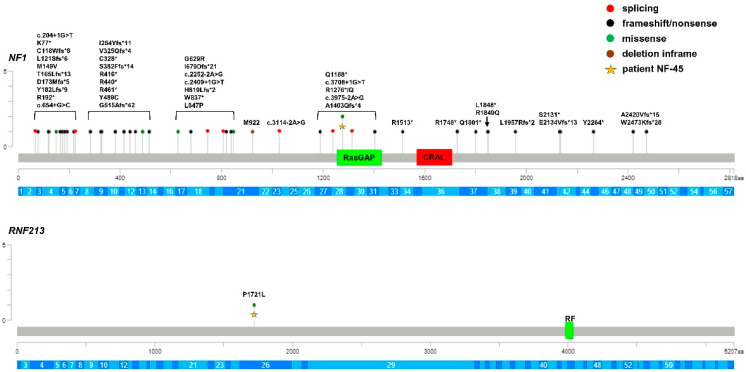
Lolliplot graph showing distribution and frequency of *NF1* and *RNF213* variants identified in NF1 patients. Yellow star identifies variants carried by the same patient. RasGAP: Ras GTPase activating protein domain; CRAL: CRAL-TRIO lipid-binding domain (cellular retinaldehyde-binding protein and TRIO guanine exchange factor); RF: RING finger domain; aa: number of protein amino acids; c.: number of nucleotide in the coding region of the gene. *: stop codon. The gene exons are schematized in dark and light blue.

**Figure 3 cancers-15-01916-f003:**
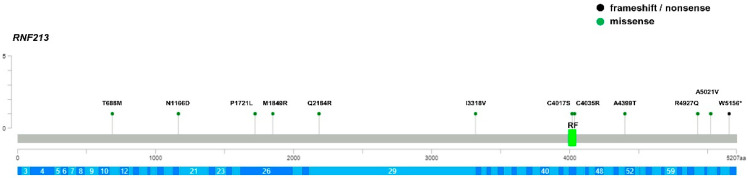
Lolliplot showing distribution and frequency of *RNF213* variants identified in MMD patients. RF: RING finger domain; aa: number of protein amino acids; * stop codon. The gene exons are schematized in dark and light blue.

**Figure 4 cancers-15-01916-f004:**
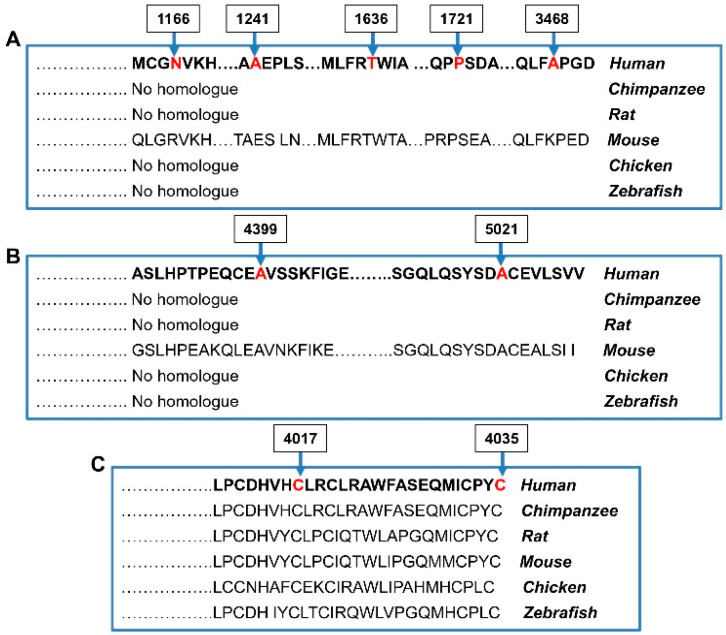
Conservation degree of *RNF213* missense variants in (**A**) N-terminal domain, (**B**) C-terminal domain and (**C**) RING finger domain, across 6 species. There is an observably high conservation of Cysteine 4017 and 4035 in the RING finger domain. The mutated amino acids identified in our study are shown in red colour.

**Table 1 cancers-15-01916-t001:** Characteristics of the 92 patients enrolled in the study.

		Overall (*n* = 92)	MMS (*n* = 20)	MMD (*n* = 25)	NF1 (*n* = 47)	*p*-Value	*p*-Value Pairwise Comparisons
Sex	Females	48 (52.2%)	12 (60.0%)	15 (60.0%)	21 (44.7%)	0.34 ^	
	Males	44 (47.8%)	8 (40.0%)	10 (40.0%)	26 (55.3%)		
Age	years	9.3 ± 5.02	11.2 ± 4.84	7.4 ± 3.33	9.5 (±5.55)	0.036 ^^*	MMD/NF1: 0.29 MMD/MMS: 0.032 * NF1/MMS: 0.56
Diagnosis age	years	5.5 ± 2.10	6.7 ± 1.93	4.6 ± 1.04	5.4 (±2.38)	0.003 ^^*	MMD/NF1: 0.18 MMD/MMS: 0.002 * NF1/MMS: 0.10
Familial cases		6	2	-	4		

MMS, Moyamoya syndrome; MMD, Moyamoya disease; NF1, Neurofibromatosis type 1; *n*, number of patients. Baseline characteristics of patients were summarized with number and percentages for categorical variables, and with mean and standard deviation for continuous variables. Pearson chi-square test (^) was performed for categorical variables and the Kruskal–Wallis test (^^) with post hoc pairwise comparison (Bonferroni correction) for continuous variables. All *p* values were two-sided and statistical significance was assumed to be at *p* < 0.05. * Statistically significant result.

**Table 2 cancers-15-01916-t002:** *NF1* and *RNF213* rare variants identified in MMS patients.

Patient	Sex	Genotype (cDNA and Protein Change)	ACMG/ClinVar/LOVD Prediction	gnomAD/ DGV Frequency	Biogear Frequency
MMS-1	F	*NF1* c.2536G>C (p.Ala846Pro) *RNF213* c.4907C>T (p.Thr1636Met)	P/P/P VUS/-/-	- 2.6 × 10^−5^	- -
MMS-2	M	*NF1* c.7285C>T (p.Arg2429 *) *RNF213* WT	P/P/P	-	-
MMS-3	F	*NF1* c.4381delA (p.Ile1461 *) *RNF213* WT	LP/-/-	-	-
MMS-4	F	*NF1* c.5483A>T (p.Asp1849Val) *RNF213* WT	P/C/VUS	-	-
MMS-5	F	*NF1* c.4276_4282delinsAG (p.Gln1447Serfs * 18) *RNF213* WT	LP/-/-	-	-
MMS-6	F	*NF1* c.2131delC (p.Arg711Alafs * 37) *RNF213* c.10403C>T (p.Ala3468Val)	P/P/- VUS/-/-	- 1.3 × 10^−4^	- 7 × 10^−4^
MMS7	F	*NF1* c.3496+1G>A *RNF213*WT	P/P/-	-	-
MMS-8	F	*NF1* c.2909delA (p.Asn970Thrfs * 3) *RNF213* WT	LP/-/-	-	-
MMS-9	M	*NF1* c.1399dupA (p.Thr467Hisfs * 3) *RNF213* WT	P/P/-	-	-
MMS-10	M	*NF1* c.5725T>C (p.Cys1930Arg) *RNF213* WT	P/C/P	-	-
MMS-11	F	*NF1* c.1642-8A>G *RNF213* WT	P/P/-	-	-
MMS-12	M	NF1c.7553_7575del *(*exon 51 del) *RNF213* WT	LP/-/-	-	-
MMS-13	F	*NF1* c5547_5943del *(*exons 38–39 del) *RNF213* WT	P/P/- -	Very rare in DGV -	-
MMS-14	F	*NF1* WT ^ *RNF213* c.3721G>A (p.Ala1241Thr)	- VUS/-/-	- 1.9 × 10^−5^	- 3 × 10^−4^
MMS-15	M	*NF1* c.6179T>G (p.Leu2081 *) *RNF213* WT	LP/-/-	-	-
MMS-16	F	*NF1* c.2356C>A (p.Gln786Lys) *RNF213* WT	LP/VUS/-	-	-
MMS-17	M	*NF1* c.610dupC (p.Leu204Profs * 12) *RNF213* WT	P/LP/-	-	-
MMS-18 ‡	M	*NF1* c.586+4dupA *RNF21*3 WT	VUS/VUS/-	-	-
MMS-19	F	*NF1* c.5010dupG (p.Tyr1671Valfs * 3) *RNF213* WT	LP/-/-	-	-
MMS-20 ‡	M	*NF1* c.5767dupC (p.Ile1923Profs * 20) *RNF213* WT	P/-/-	-	-

LOVD, Leiden open Variation Database; DGV, Database of Genomic Variants; Biogear, in-house exome controls dataset; M, male; F, female; del, deletion; VUS, variant of uncertain significance; P, pathogenic; LP, likely pathogenic; C, conflicting; WT, wild type; * stop codon; ‡, familial case; ^ The patient was not investigated by MLPA and cDNA sequencing. Reference sequences: *NF1*: NM_000267.3; *RNF213*: NM_001256071.3.

**Table 3 cancers-15-01916-t003:** *NF1* and *RNF213* rare variants identified in NF1 patients.

Patient	Sex	Genotype (cDNA and Protein Change)	ACMG/ClinVar/LOVD Prediction	gnomAD Frequency	Biogear Frequency
NF-1	M	*NF1* c.1885G>A (p.Gly629Arg) *RNF213* WT	P/P/P	-	-
NF-2	M	*NF1* c.5242C>T (p.Arg1748 *) *RNF213* WT	P/P/P	-	-
NF-3	M	*NF1* c.2970_2972del (p.Met992del) *RNF213* WT	P/P/P	8.2 × 10^−6^	-
NF-4	M	*NF1* c.545del (p.Tyr182Leufs * 9) *RNF213* WT	LP/-/-	-	-
NF-5	M	*NF1* c.493del (p.Thr165Leufs * 13) *RNF213* WT	LP/-/-	-	-
NF-6	M	*NF1* c.1544_1545del (p.Gly515Alafs * 42) *RNF213* WT	LP/-/-	-	-
NF-7	M	*NF1* c.973_974del (p.Val325Glnfs * 4) *RNF213* WT	LP/-/-	-	-
NF-8	F	*NF1* c.360dup (p.Leu121Serfs * 6) *RNF213* WT	LP/-/-	-	-
NF-9	F	*NF1* c.1144dup (p.Ser382Phefs * 14) *RNF213* WT	LP/-/-	-	-
NF-10	F	*NF1* c.5870del (p.Leu1957Argfs * 2) *RNF213* WT	LP/-/-	-	-
NF-11	F	*NF1* c.6391_6394del (p.Ser2131 *) *RNF213* WT	LP/-/-	-	-
NF-12 ‡	M	*NF1* c.850del (p.Ile284Tyrfs * 11) *RNF213* WT	LP/-/-	-	-
NF-13 ‡	M	*NF1* c.656del (p.Ala219Aspfs * 6) *RNF213* WT	LP/-/-	-	-
NF-14	M	*NF1* c.5401C>T (p.Gln1801 *) *RNF213* WT	P/P/P	-	-
NF-15	F	*NF1* c.229A>T (p.Lys77 *) *RNF213* WT	LP/-/-	-	-
NF-16	M	*NF1* c.3562C>T (p.Gln1188 *) *RNF213* WT	P/P/P	-	-
NF-17	F	*NF1* c.2033dup (p.Ile679Aspfs * 21) *RNF213* WT	P/P/P	-	-
NF-18	M	*NF1* c.3114-2A>G *RNF213* WT	P/LP/P	-	-
NF-19	M	*NF1* c.1381C>T (p.Arg461 *) *RNF213* WT	P/P/P	-	-
NF-20 ‡	F	*NF1* c.354_356delinsG (p.Cys118Trpfs * 8) *RNF213* WT	LP/-/-	-	-
NF-21	M	*NF1* c.3975-2A>G *RNF213* WT	P/P/P	-	-
NF-22	F	*NF1* c.5546G>A (p.Arg1849Gln) *RNF213* WT	P/P/P	-	-
NF-23	F	*NF1* c.4537C>T (p.Arg1513 *) *RNF213* WT	P/C/P	6 × 10^−6^	-
NF-24	F	*NF1* c.574C>T (p.Arg192 *) *RNF213* WT	P/C/P	1 × 10^−5^	-
NF-25	M	*NF1* c.7259-1_7259del (p.Ala2420Valfs * 15) *RNF213* WT	LP/-/-	-	-
NF-26	F	*NF1* c.2511G>A (p.Trp837 *) *RNF213* WT	P/P/P	-	-
NF-27	F	*NF1* c.3826C>T (p.Arg1276 *) *RNF213* WT	P/P/P	-	-
NF-28	M	*NF1* c.2409+1G>T *RNF213* WT	P/P/P	-	-
NF-29	M	*NF1* c.6401_6402del (p.Glu2134Valfs * 13) *RNF213* WT	LP/-/-	-	-
NF-30	M	*NF1* c.1246C>T (p.Arg416 *) *RNF213* WT	P/P/P	8 × 10^−6^	-
NF-31	M	*NF1* c.2540T>C (p.Leu847Pro) *RNF213* WT	P/C/P	-	-
NF-32 ‡	F	*NF1* c.3708+1G>T *RNF213* WT	P/LP/P	-	-
NF-33	F	*NF1* c.4206del (p.Ala1403Glnfs * 4) *RNF213* WT	LP/-/-	-	-
NF-34	M	*NF1* c.983_984del (p.Cys328 *) *RNF213* WT	P/P/P	-	-
NF-35	M	*NF1* c.516delT (p.Asp173Metfs * 5) *RNF213* WT	LP/-/-	-	-
NF-36	M	*NF1* c.2540T>C (p.Leu847Pro) *RNF213* WT	P/C/P	-	-
NF-37	F	*NF1* c.2456del (p.His819Leufs * 2) *RNF213* WT	LP/-/-	-	
NF-38	F	*NF1* c.6792C>A (p.Tyr2264 *) *RNF213* WT	P/P/P	-	-
NF-39	M	*NF1* c.1318C>T (p.Arg440 *) *RNF213* WT	P/P/P	-	1 × 10^−4^
NF-40	M	*NF1* c.7417_7425delinsAAGGT(Trp2473Lysfs * 28) *RNF213* WT	LP/-/-	-	-
NF-41	F	*NF1* c.2252-2A>G *RNF213* WT	P/C/P	-	-
NF-42	F	*NF1* c.204+1G>T *RNF213* WT	P/P/P	6 × 10^−6^	-
NF-43	F	*NF1* c.3445A>G (p.Met1149Val) *RNF213* WT	P/C/P	6 × 10^−6^	-
NF-44	F	*NF1* c.1466A>G (p.Tyr489Cys) *RNF213* WT	P/P/P	6 × 10^−6^	-
NF-45	M	*NF1* c.3827G>A (p.Arg1276Gln) *RNF213* c.5162C>T (p.Pro1721Leu)	P/P/P VUS/-/-	1 × 10^−5^ 1 × 10^−3^	- 2.7 × 10^−3^
NF-46	F	*NF1* c.5543T>G (p.Leu1848 *) *RNF213* WT	P/P/P	-	-
NF-47	M	*NF1* c.654+1G>C *RNF213* WT	LP/-/P	-	-

M, male; F, female; LOVD, Leiden open Variation Database; Biogear, in-house exome controls dataset; VUS, variant of uncertain significance; P, pathogenic; LP, likely pathogenic. C, conflicting; * stop codon; ‡ Familial case. Reference sequences: *NF1*: NM_000267.3; *RNF213*: NM_001256071.3.

**Table 4 cancers-15-01916-t004:** *RNF213* genotype of MMD patients.

Patient	Sex	*RNF213* Genotype	Protein Domain	ACMG/ClinVar/LOVD Prediction	gnomAD Frequency	Biogear Frequency
MMD-1	F	c.5546T>G (pMet1849Arg)	N-term	VUS/-/-	-	-
MMD-2 ^†^	F	**c.15062C>T (p.Ala5021Val)**	C-term	VUS/B/-	1.5 × 10^−4^	-
MMD-3 **^‡^**	M	c.10443G>A (p.Ala3481=)		B/-/-	-	7 × 10^−4^
MMD-4	F	c.4146T>G (p.Thr1382=)		B/B/-	8 × 10^−3^	5 × 10^−3^
MMD-5	M	WT		-		
MMD-6	M	**c.12050G>C (p.Cys4017Ser)**	RF	VUS/-/-	-	-
MMD-7	F	**c.5162C>T (p.Pro1721Leu)**	N-term	VUS/VUS/-	1 × 10^−3^	1 × 10^−3^
MMD-8	F	WT		-		
MMD-9	F	WT		-		
MMD-10	M	WT		-		
MMD-11	M	c.12103T>C (p.Cys4035Arg)	RF	VUS/-/-	-	-
MMD-12	M	WT		-		
MMD-13	F	WT		-		
MMD-14	F	**c.9952A>G (p.Ile3318Val)**	N-term	B/B/-	1 × 10^−3^	1 × 10^−2^
MMD-15	F	c.2063C>T (p.Thr688Met)	N-term	B/B/-	1 × 10^−3^	3 × 10^−4^
MMD-16	F	c.3496A>G (p.Asn1166Asp)	N-term	VUS/-/-	-	-
MMD-17	M	WT		-		
MMD-18	F	WT		-		
MMD-19	M	c.6042C>T (p.Asn2014=) c.15467G>A (p.Trp5156 *)	- C-term	B/B/- LP/-/-	1 × 10^−3^ -	3 × 10^−3^ -
MMD-20	F	c.14780G>A (p.Arg4927Gln)	C-term	VUS/-/-	5 × 10^−5^	-
MMD-21	F	WT		-		
MMD-22	F	WT		-		
MMD-23	M	WT		-		
MMD-24	F	**c.6551A>G (p.Gln2184Arg)**	N-term	B/LB/-	2 × 10^−3^	-
MMD-25	M	**c.13195G>A (p.Ala4399Thr)**	C-term	B/P/-	8 × 10^−3^	1 × 10^−3^

M, male; F, female; LOVD, Leiden open Variation Database; Biogear, in–house exome controls dataset; X, STOP codon; VUS, variant of uncertain significance; B, benign; LP, likely pathogenic; WT, wild type; ^†^ Asiatic origin; ^‡^ African origin; * stop codon; RF, RING finger domain; Bold character, variants just identified in MMD patients; C-term, C-terminal region of the protein; N-term, N-terminal region of the protein. Reference sequence: *RNF213*: NM_001256071.3.

**Table 5 cancers-15-01916-t005:** Putatively damaging *RNF213* rare variants burden tests.

	Carrier Frequency				IRR (95%CI); *p*-Value	
Domain	MMS (*n* = 20)	MMD (*n* = 25)	Biogear (*n* = 2504)	gnomAD (*n* = 141,456)	MMS	MMD
N-term	3 (15%)	3 (12%)	282 (11%)	NA	1.33 (0.273–3.932); *p* = 0.58 * NA **	1.07 (0.22–3.15); *p* = 0.85 * NA **
RING finger	0	2 (8%)	8 (0.3%)	227 (0.1%)	0.00 (0.00–73.05); *p* = 0.80 * 0.00 (0.00–115.88); *p* = 0.86 **	25.04(2.59–125.47); *p* = 0.004 * 49.85 (6.00–182.17); *p* < 0.001 **
C-term	0	4 (16%)	61 (2.3%)	1373 (0.9%)	0.00 (0.00–7.80); *p* = 0.49 * 0.00 (0.00–19.03); *p* = 0.66 **	6.57(1.73–17.69); *p* = 0.004 * 16.48 (4.49–42.30); *p* < 0.001 **
RING finger + C-term	0	6 (24%)	69 (2.7%)	1600 (1%)	0.00 (0.00–6.88); *p* = 0.46 * 0.00 (0.00–19.03); *p* = 0.66 **	8.71(3.09–19.94); *p* < 0.001 * 21.22 (7.78–46.29); *p* < 0.001 **

Biogear, in house-exome controls; IRR, incidence rate ratio; CI, confidence interval; NA: data not available. * comparisons made versus Biogear dataset; ** comparisons made versus gnomAD dataset. Comparison of two rates (sample vs. healthy databases) was expressed as ratio of the two rates (IRR: incidence rate ratio) with its 95% confidence interval (calculated with exact Poisson method) and associated *p* value. Statistical significance was assumed at *p* < 0.05.

## Data Availability

The data that support the findings of this study are available in the article.

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
