# Peer review of "Moyamoya Vasculopathy in Neurofibromatosis Type 1 Pediatric Patients: The Role of Rare Variants of RNF213"

_cancers, 2023, doi:10.3390/cancers15061916_

Round 1

Reviewer 1 Report

Reviewer opinion

Neurofibromatosis type 1 is one of the neurocutaneous disorders characterized by skin manifestations and neurofibromas. Additionally, individuals with NF1 may present with variable features resulting from the maldevelopment of neuroectodermal tissues during embryonic development. Moyamoya disease (MMD)-like syndrome is a rare association symptom observed in some NF1 patients due to vascular defects. MMD has a genetic predisposition, and some MMD patients have a positive family history of the condition. Recently, a few sequence variants, including the p.R4810K missense variant in the RNF213 gene, have been identified as being associated with MMD. This variant is considered to be a disease-causing mutation that has a founder effect in Asian populations. This study aims to identify any rare sequence variants in the RNF213 gene that may be associated with MMD susceptibility in individuals with NF1. The opinions of the reviewer are listed in the following paragraphs.

Major concerns,

1.      The paper primarily showcased several genetic sequencing variants discovered in three patient cohorts: those with only NF1, those with both NF1 and MMS, and those with MMD only. To investigate a possible link between the RNF213 sequence variant and MMS susceptibility in NF1 patients, the PI assessed the relative risk of the variant among the three cohorts. However, the study's main takeaway is that the allele frequency in the disease cohort was too low, rendering the findings inconclusive. To identify a modifier gene, a sufficiently large NF1 cohort with and without MMS symptoms is necessary to increase the statistical power. Relying on data from only 20 patients with both MMS and NF1, compared to 47 patients with only NF1, would be risky when drawing a conclusion. Needless to say, there are only three in these twenty patients who harbor a sequence variant in RNF213 gene.  

2.      The study found that 94% of the patients had a Caucasian ancestry. However, to identify any variants in the RNF213 gene that contribute to the phenotype of MMS in NF1 patients, it is important to recruit individuals with diverse ethnic backgrounds. Otherwise, a private, rare sequence variant in RNF213 might be a confounding factor, leading to false-positive results.

3.      While there are multiple factors that contribute to the phenotype of MMS, confirming the genetic heritability that leads to the symptoms of MMS can be challenging.

4.      There is a bimodal increase in the age of onset among patients affected by MMD. Based on Table 1, the age of diagnosis for some patients is younger than 10 years, and the oldest patient enrolled was 18 years old. While a few patients experienced cerebral hemorrhage in their second or third decade of life, it may be premature to consider affected symptoms in pediatric patients as a confounding factor at this stage.

A few minor suggestions:

1.      The seventh line of third paragraph in Introduction section (Page 2): 0.5-1.5 per 100.000 should be 0.5-1.5 per 100,000 and, 0.1 per 100.000 should be 0.1 per 100,000

2.      The eleventh line of third paragraph in Introduction section (Page 2): …, sensory and consciousness disturbance, …….., movement disorder………. à …., sensory and consciousness disturbances, …., movement disorders ……..

3.      The thirteenth line of third paragraph in Introduction section (Page 2): ….. and the treatment strategy include …….. à …. and the treatment strategy includes ….

4.      The last line of third paragraph in Introduction section (Page 2): …choice of….. à ….selection of….

5.      In the section of “Patients and controls”: The document from the internal review board needs to be presented.

6.      In the ‘Next generation sequencing’ section: Could the amplicon size of 375 bp be too long for the short-read sequencing machine, considering that the extreme end region might be difficult to read?

7.      In the “MLPA” analysis: Has testing for exon deletion or duplication in the RNF213 gene been conducted?

8.      The first four paragraphs of the discussion section sound like an introduction or offer background details, resulting in numerous sentences that could be condensed for greater conciseness.

Author Response

Major concerns

-We agree that a larger cohort of patients is necessary to confirm our negative association between RNF213 variants and increased risk of Moyamoya vasculopathy in NF1 patients. As we described in “Methods” section, our selection of MMS cases was made on the basis of an internal database of 600 patients described in Scala et al. (2021). The prevalence of Moyamoya in NF1 patients is estimated at most at 4-6%, thus, we should have recruited 24-36 patients. Since most cases are asymptomatic, our series of 20 MMS patients is coherent and represents a great effort for a single-centre study. In fact, all series published since 1976 reported few patients and, overall, fewer than 100 patients with MMS-NF1 have been described in the literature (Vargiami et al. 2014). Very recently, Po et al. (2022) published a multicentric Italian pediatric cohort of 65 cases, but only 6 have Neurofibromatosis 1, and only 1 molecularly proven. Thus, given the rarity of NF1 and Moyamoya association, it is difficult to carry out studies with sufficient statistical power. Our study aims to give a contribution, but it will have to be confirmed by other studies in reply. Phi et al. (2016) investigated the role of RNF213 c.14576G>A variant in MMS pathogenesis of East Asian population and they investigated small cohorts of 16 MMS patients and 97 NF1 patients without MMS as controls. Furthemore, to the best of our knowledge, our study represents the first genetic study on Moyamoya- NF1 patients of Caucasian origin, and even if the conclusions are weak, they may represent the start point for additional studies investigating the NF1-Moyamoya connection.-Although Moyamoya is 10 times more frequent among Asiatic than in European and NF1 has a stable prevalence worldwide, the association between Moyamoya and NF1, typical of pediatric age, is more frequent in Caucasian individuals, demonstrating that different genetic backgrounds among ethnic groups may play a role. Our study aims to investigate the genetic landscape of MMS in patients of European ancestry. We are conscious that rare variants tend to be more geographically clustered since they are of recent origin and that population stratification, due to different ancestries, has been shown to be a confounding factor that could lead to false positive in rare variant association studies. However, we selected controls from our in–house exomes dataset who have similar genetic background, although different genotyping and sequencing platforms have different genotyping qualities and error rates and this approach should be used with extreme caution. -We agree that multiple factors may contribute to the phenotype of MMS. Here we showed that RNF213 has, if any, only a minor role, and other genes that can be implicated could be discovered with different approaches such as whole exomes sequencing, which is what we are doing. All MMS/MMD patients enrolled in the study manifested severe symptoms (ischemic attacks) and all underwent neurosurgery treatment consisting of indirect bypass re-vascularization. Moreover, Moyamoya diagnosis was supported by brain MRI or cerebral angiography . Thus, it is unlikely that symptoms of patients are a confounding factor.

Minor concerns

-We corrected all typos.

- Ethical review and approval was not due since the genetic studies had a diagnostic purpose, and the explicitly identifying information concerning subjects were removed or encrypted.

- IonTorrent/Thermofisher GeneStudio S5 platform is able to sequence amplicons sizes up to 400 bp using an appropriate number of flows to sequence. The PCR primers were designed by on line tool of Thermo Fisher Scientific, the Ion AmpliSeq Designer (http://www.ampliseq.com).

-Although MRC Holland continues to innovate with the development of new MLPA assays, to date an assay for RNF213 has not been developed yet. Furthemore, copy number variations have not been reported for RNF213 gene.

- We condensed the first four paragraphs of the discussion as the reviewer suggest.

Reviewer 2 Report

The article by Ognibene et al, discusses the association between neurofibromatosis type 1 (NF1) and Moyamoya syndrome (MMS), which is characterized by the occlusion of intracranial arteries and increased risk of ischemic and hemorrhagic events. The study investigated the role of RNF213 gene mutations as genetic modifiers of MMS phenotype in a cohort of pediatric patients. The results showed that RNF213 gene mutations are involved only in the occurrence of Moyamoya disease (MMD), and not in the increased risk of MMS in NF1 patients. The study also suggests that the loss of neurofibromin 1 alone could be responsible for the excessive proliferation of vascular smooth muscle cells, causing Moyamoya arteriopathy associated with NF1. The findings of the study highlight the importance of further research to elucidate the possible role of other genes and enhance our understanding of the pathogenesis and treatment of MMS. Overall, the study provides important insights into the genetic mechanisms underlying the development of MMS and its association with NF1.

While the article provides valuable insights into the association between NF1 and MMS, I have a few comments that could improve the manuscript. Firstly, the introduction could be expanded to provide a more detailed explanation of NF1 and MMS for readers who may not be familiar with these conditions. Additionally, the results section could benefit from a more detailed description of the results used in the study, as well as a more comprehensive presentation of the results, including replacing tables by figures as detailed bellow. Finally, the authors may consider discussing the clinical implications of the study's findings, including potential implications for the diagnosis and treatment of MMS in NF1 patients. Overall, these suggestions could help to improve the clarity and impact of the manuscript.

Author Response

Introduction: we accepted all the sentences the reviewer introduced and we added the corresponding references.

Methods: we corrected all typos.

Results:

-We deteleted Figure 1 and we introduced 3 new figures (lolliplot graphs) and we replaced the old Figure 2 with the new Figure 4.

-We inserted in Table 1 the number of familial forms of Neurofibromatosis.

-We explained the meaning of the p-values in Table 1: the value p=0.036 was calculated when we compared overall differences of age among groups; we added a pairwise comparison that evidences a significant difference among MMD and MMS group. The value p=0.003 was calculated for overall comparison of age at diagnosis. We found a significant lower age at diagnosis between MMD and MMS group. We specified in the legend the type of statistical test.

-We have made tables legends more comprehensive and we presented besides Tables 2, 3 and 4 the lolliplots graphs.

-De novo RNF213 variants localized in the RING finger domain becomes Figure 4. We deleted panel A and B as you requested and we replaced them with studies on conservation degree of other RNF213 missense variants.

Discussion

-As you suggested we added in the text that beside RNF213, other genes have been associated with Moyamoya syndrome including ACTA2, GUCY1A3, and MYH11, among others.

-We discussed the results in the context of MMD, MMS and NF1 incidence in populations of different origins. We added the following sentence: “Thus, while in East Asian countries MMD is strongly associated to the p.R4810K variant in the RNF213 gene that segregates with disease in an autosomal dominant manner with reduced penetrance, in accordance with findings of Guey et al. (2017), our results provided evidences that several heterozygous rare RNF213 variants, non p.R4810K, localized in the C-terminal hot spot domain play a role for a substantial proportion of non syndromic Moyamoya patients of European ancestry”.

-We discussed the clinical implications of the study's findings, including potential implications for the diagnosis and treatment of MMS in NF1 patients. We added the following sentence: “In a diagnostic context, our study raises the question of the putative benefit of a genetic screening of RNF213 in MMS patients. In fact, MMD is a progressive disease and prevention of strokes and the resulting co-morbidities depends on the early identification of at risk individuals. In general, there is no consensus about the management of NF1-associated vasculopathies and pediatric care guidelines do not recommend neuroimaging as a surveillance modality. Thus, the identification of non invasive prognostic factors is crucial for management of pediatric patients. The data presented here suggested that screening for rare variants in the C-terminal domain of RNF213 should be considered only for pediatric patients with isolated MMD and not for syndromic MMS patients”.

Reviewer 3 Report

This is a unique and interesting study to explore RNF213 rare variants in MMD. While the patients number is always a limit for the rare disesase study, this work is trying to exploring potential factors in RNF213 gene related to the MMS and MMD with and without NF1 background.  The conclusion provides modest prognostic benefit using RNF213 variants.

Many typos and mislabeling in the manuscript need to improve or correct 

1) "01. per 100.000"

2) Two "Figure 1" 

3) * in Figure 1 is confusing. What does the * mean on p-value? What type of test was used?

4) All types of statistical tests should be clearly stated under all tables and figures. 

5) The primer set used for NF1 and RNF213 sequencing should be disclosed as supplementary data. 

6) Please search and confirm all the hazard NF1 variants in this study via the database (https://databases.lovd.nl/shared/variants/NF1/ ) and specify which databases are used in this study.

7) The in-house Biogear database should be cited or described. Since it mainly collect the caucasian data, and RNF213 has low prevalence in Caucasian, the false positive could be high. Additional public database should be used. 

Author Response

-We corrected in the text: MAF<1:10.000

-We deleted Figure 1 (as also suggested by the Reviewer 2) and we added three new figures (suggested from the Reviewer 2) and a revised version of Figure 2 that become Figure 4 showing studies on conservation degree of other RNF213 missense variants.

-Since we deleted Figure 1, we specified in the legend of Table 1 the statistical tests performed for discrete and continue variables.

-We specified in legends of Table 1 and Table 5 the type of the statistical test.

-We provided sequences of primers used for NF1 cDNA re-sequencing in “Supplementary data”. Primers set used for NGS re-sequencing were designed with the Ion AmpliSeq Designer tool of by Thermo Fisher Scientific, therefore the sequence is their property.

-We checked the presence of the NF1 and RNF213 variants in LOVD and we specified in the Methods section that we checked two public databases : ClinVar and LOVD.

-Biogear dataset, the Gaslini in-house exomes dataset, will be the subject of a publication by its curator, the statistician Dr. Uva P., who is one of the co-authors. We specified in the text that this dataset collected sequencing files and phenotype data from families whose probands are children with neurodevelopmental disorders and individuals used for statistical tests are unaffected parents with ethnically-matched (Caucasian) origin. We used also gnomAD, a resource that collected 125,748 exome sequences from >60 distinct populations from Africa, Europe, the Middle East, South and Central and South Asia, East Asia, Oceania, and the Americas.

Round 2

Reviewer 1 Report

The modified manuscript has shed light on certain aspects that were pertinent to my main concern. I do not have any additional comments.